# Monitoring Shear Behavior of Prestressed Concrete Bridge Girders Using Acoustic Emission and Digital Image Correlation

**DOI:** 10.3390/s20195622

**Published:** 2020-10-01

**Authors:** Fengqiao Zhang, Gabriela I. Zarate Garnica, Yuguang Yang, Eva Lantsoght, Henk Sliedrecht

**Affiliations:** 1Department of Engineering Structures, Delft University of Technology, 2628 CN Delft, The Netherlands; G.I.ZarateGarnica@tudelft.nl (G.I.Z.G.); Yuguang.Yang@tudelft.nl (Y.Y.); E.O.L.Lantsoght@tudelft.nl (E.L.); 2Politécnico, Universidad San Francisco de Quito, Quito EC 17015, Ecuador; 3Rijkswaterstaat, Ministry of Infrastructure and Water Management, 3526 LA Utrecht, The Netherlands; henk.sliedrecht@rws.nl

**Keywords:** acoustic emission measurements, crack identification, cracking, digital image correlation, prestressed concrete bridge girders, shear

## Abstract

In the Netherlands, many prestressed concrete bridge girders are found to have insufficient shear–tension capacity. We tested four girders taken from a demolished bridge and instrumented these with traditional displacement sensors and acoustic emission (AE) sensors, and used cameras for digital image correlation (DIC). The results show that AE can detect cracking before the traditional displacement sensors, and DIC can identify the cracks with detailed crack kinematics. Both AE and DIC methods provide additional information for the structural analysis, as compared to the conventional measurements: more accurate cracking load, the contribution of aggregate interlock, and the angle of the compression field. These results suggest that both AE and DIC are suitable options that warrant further research on their use in lab tests and field testing of prestressed bridges.

## 1. Introduction

Many bridges in the Netherlands were built in the 1960s and 1970s. These bridges were designed for lower live loads using past code provisions, which may lead to insufficient capacities when assessed with the current code provisions. Upon assessment, many prestressed concrete bridge girders are reported to have insufficient shear–tension capacity [1,2] according to NEN-EN 1992-1-1:2005 [3]. To investigate the actual shear behavior of the existing prestressed concrete girders, four girders were taken from a typical demolished concrete girder bridge (the Helperzoom bridge) in the Netherlands. Shear tests were performed on these specimens in the Stevin II Laboratory at Delft University of Technology. The experimental program was reported in [4], which proved additional capacity compared to the code provisions [5]. In addition to the conventional measurement techniques, we carried out Acoustic Emission (AE) measurements and photographs were captured during the shear tests for Digital Image Correlation (DIC). These measurements provided additional insights into the structural behavior during the cracking and shear failure process.

AE represents the elastic waves excited by sudden changes in the concrete, such as cracking. By continuously recording and processing the AE signals, one can detect the damage process during loading and unloading. An AE event includes a group of signals that come from one source. With sufficient accuracy [6], one can find the crack profile by locating the source of AE events based on the arrival times of signals, which is called AE source localization [7]. Besides, features of AE signals can help classify different sources of AE events, which is called AE source classification [8,9,10]. Both AE source localization and classification are implemented in this paper.

DIC is an optical technique for the evaluation of displacements and strains. It is based on comparison between images taken during the loading of a specimen. The advantages are that DIC is a non-contact measuring technique and provides full-field displacement measurements of an entire specimen surface. Thus, DIC has been used to investigate the shear-carrying mechanisms in concrete members without shear reinforcement [11,12,13], and prestressed concrete structures [14,15], in the laboratory. However, the application to full-scale concrete elements is still rare [16,17,18].

This paper interprets the insights from AE and DIC measurements with regard to three elements that are considered important for understanding the cracking and shear failure process of prestressed concrete girders, namely: cracking load, aggregate interlock, and angle of compression field.

The cracking load is important as it can help estimate the working prestressing level of the girder [19] which influences the strain distribution in the cross-section and the deflections. With displacement-based measurements, the cracking load could only be detected by checking the change in stiffness in the load-deflection curve [20]. As cracking is a local phenomenon, which has limited effect on the global stiffness, the traditional approach is not sensitive. AE, on the other hand, is known to be sensitive to detecting local (micro)cracking, thus giving a more accurate estimation of the cracking load and thus the prestressing level.

The study of the contribution of aggregate interlock and the angle of the compression field can help understand the shear-transfer mechanisms of prestressed concrete girders. Aggregate interlock and stirrups are considered as the main contributors to the shear strength of a cracked concrete girder based on the Modified Compression Field Theory (MCFT) [21], which is the theoretical basis of the AASHTO (American Association of State Highway and Transportation Officials) shear provisions [22], and the *fib* Model Code 2010 [23]. According to the aggregate interlock theory of Walraven [24], the shear force transfer across cracks through aggregate interlock can be determined once the relative displacement across the crack trajectory is determined. With the DIC full-field displacement measurement, we can determine the displacement at any point along the crack to estimate the aggregate interlock and the tensile forces in the stirrups locally.

In plasticity-based models, such as the Variable Angle Truss Model (VATM) [25], a compression field is typically used to model the transfer of shear forces in the cracked concrete web of the girder. Such model is employed in NEN-EN 1992-1-1:2005 [3], and is widely accepted in the design of shear-reinforced concrete elements. This approach idealizes the shear transfer in a cracked member to a truss model, which has diagonal concrete struts in compression with a variable angle (21.8° ≤ *θ* ≤ 45°). In existing prestressed concrete girders, limited shear reinforcement was applied according to the design provisions of that time. As a result, the minimum truss angle defined by the current code is often reached for these elements. However, the actual angle of the compression field appears to be lower than the minimum allowable angle specified by the code provision [5]. DIC provides the possibility to measure the angle of the compression filed without dense installation of traditional displacement measurements, such as LVDTs (linear variable differential transformers).

The significance of the presented research lies in the additional insights that AE and DIC can provide in the monitoring of full-scale specimens. This work gives a better understanding of the cracking and shear-carrying behavior of girders in slab-between-girder bridges, and provides recommendations for AE and DIC measurements on bridges in the field.

## 2. Materials and Methods

### 2.1. Description of Experiments

This section presents the relevant information from the experiments. Additional information on this series of experiments can be found in a companion paper [4].

#### 2.1.1. Materials and Geometry

The average concrete compressive strength was 76.3 MPa as determined on cores taken from the Helperzoom viaduct, and the splitting tensile strength was 5.4 MPa. The Young’s modulus of the concrete was 39,548 MPa.

The mild steel (stirrups and longitudinal bars) had a yield strength of 454 MPa and an ultimate strength of 655 MPa, as determined on material samples in the laboratory. The stirrup ratio was 0.196%.

Samples of the prestressing steel were also tested in the lab. The prestressing steel reached a strain of 0.01 at 1433 MPa and failed at 1824 MPa with a strain of 0.0535. Ten draped tendons with 12 strands of 7 mm diameter were used, of which three tendons were anchored in the top flange and the remaining tendons in the anchorage block.

The total length of the girders was 23.4 m. The girders were cut to about half of their length, varying between 10.51 m and 12.88 m. The resulting four girder segments were numbered consecutively as HPZ01 through HPZ04.

#### 2.1.2. Test Setup

Figure 1a shows an overview of the test setup, including the girder, the steel frame used for applying the loading, one of the supports, and the actuator. The girders were simply supported with a span of 9.6 m. A single concentrated load was applied through a loading plate of 300 mm × 300 mm. The center of the load was at 2903 mm from the center of the support in HPZ01 and HPZ02, and at 4400 mm in HPZ03 and HPZ04. The purpose was to investigate the influence of load position on shear capacity. Detailed analysis can be found in a companion paper [4]. To avoid failure on the unwanted side, slipping of the prestressing strands and a possible spalling failure during the test, vertical prestressing was applied near the other support.

The load was applied in cycles of loading and unloading to predefined load levels, instead of monotonically to failure (Figure 2). For each beam, four load levels were chosen based on the expected structural responses during the test. At the first load level, no cracks occurred. By the second load level, a first flexural crack was expected. By the third load level shear cracks were expected. After the fourth load level, the specimen failed. The first three load levels are composed of three load cycles each. Small load steps of 50 kN were taken when loading to the load level for the first time. With these load cycles, crack opening and closing can be followed.

#### 2.1.3. Traditional Instrumentation

In this paper, traditional instrumentation included LVDTs, laser distance finders and a load cell. LVDTs were applied on one face of the girder in the web to follow the development of deformations (in horizontal, vertical, and diagonal directions). Laser distance finders were installed at the position of the load to measure the deflection of the girder. A load cell was installed at the hydraulic jack to measure the applied load.

In each experiment, similar sensor layout was used to cover the interesting measuring area. Figure 3 shows the sensor layout as used for HPZ04, including LVDTs and the AE sensors which are discussed in the next section. The other face of the girder was used for the DIC measurements, as also shown in Figure 1a.

### 2.2. AE Monitoring

In this paper, AE sensors were installed on the same side as the LVDTs in the web and on the bottom surface. The sensors were resonant type sensors with center frequencies of 60 kHz, i.e., R6I, from MISTRAS Group, Inc. [26]. They were coupled with the concrete surface using electricity compound, and fixed by metal holders, shown in Figure 4a. The sensors on the bottom surface were installed 175 mm inside from the outer edge of the bottom flange, and on the same vertical 2D plane as other sensors in the web (see Figure 3). The area enclosed by AE sensors was referred to as the measuring area, which may vary slightly in the four experiments.

For AE monitoring of such large-scale specimens, the sensor spacing needs to be controlled. Using a large sensor spacing can increase the measuring area, but may decrease the source localization accuracy [6] or falsely classify tensile cracking into friction [27,28]. This paper controlled the sensor spacing to be 0.5 m and 0.52 m in horizontal and vertical directions, respectively, as shown in Figure 3, which was proven to be effective in the previous measurements on large-scale reinforced concrete structures [29]. To cover the relevant area in shear span, and controlling the sensor spacing, 13, 13, 15 and 20 AE sensors were applied in HPZ01-04, respectively.

#### 2.2.1. Measurement of Wave Transfer Properties of Concrete Medium

Before testing, measurement of the wave transfer properties of the concrete medium were carried out on the uncracked girder Figure 4a. The measurement followed the same procedure described in [30]. During wave transfer, amplitude drop generally came from two sources: the material attenuation and the spreading loss [31]. Here, the amplitude drop was expressed in decibel value of the ratio of the source signal amplitude and the received signal amplitude. The function is a sum of a linear function (from material attenuation, with the material attenuation factor as the slope) and a logarithmic function (from spreading loss) to the wave travel distance. The mathematical expressions can be found in [6]. By fitting the amplitude drop function to the measured data which were acquired at different travel distances (Figure 4b), the material attenuation factor was found to be 8 dB/m. In total, an amplitude drop of 26 dB was found at 1 m from the source, including material attenuation of 8 dB and spreading loss of 18 dB. In our tests, the noise level was measured to be less than 40 dB. Thus, a source signal of 66 dB can be detected by sensors within 1 m away. Therefore, the current maximum sensor spacing of 0.52 m was acceptable. In the baseline measurement, another important material property, i.e., wave speed, was determined as the ratio of distance and travel time (4670 m/s), see Figure 4c. This value was used for the source localization in Section 2.2.2.

#### 2.2.2. Source Localization

One of the important parameters in AE source localization is the arrival time of a signal at the receiver. To determine the arrival time, this paper used the threshold method, which determined the arrival time as the first point of the signal envelope that crossed the pre-defined threshold level (Figure 5a). A fixed threshold level of 40 dB was taken to determine the arrival time. The difference between the determined and the real arrival times was referred to as the arrival time picking error. Though other advanced methods based on the waveform can provide a smaller picking error [32], the threshold method is still widely applied, as it is time-efficient, especially in dealing with a large amount of data in real-time. Therefore, it is chosen in this study.

Based on the recorded signal arrival times, grid search method was used to estimate the source location [33]. The grid point, of which the calculated arrival times at the sensors were best matched with the recorded arrival times, was determined as the source location [7]. In this paper, the measuring area was discretized into a grid with size 5 × 5 mm. The mathematical formulation used in source localization can be found in our previous paper [6].

For existing concrete structures, inconsistencies like the presence of cracks (from the concrete structure) and the arrival time picking error (from the arrival time picking algorithm) will result in an error on the estimated source location. The distance between the actual source location and the estimated location is called the source localization error. A previous study showed an error less than 15 cm for concrete structures when four AE sensors were used [6]. The source localization error can be reduced when more sensors receive signals from the same source.

To quantify the spatial distribution of the located AE events, we divided the measuring area into cells, and counted the number of AE events located in each cell, which was referred to as local cumulative AE events. This approach has been described in detail in [29]. The cell dimension is limited by the source localization error, and was taken as 15 × 15 cm in this paper. The approach of calculating the local cumulative AE events is illustrated in Figure 6.

Local cumulative AE events within each load step was calculated. The start and end point of each load step are marked in Figure 2. In this way, we can track the change of local cumulative AE events with increasing load. As the number of AE events is related to the cracking activities [34], the change of local cumulative AE events was expected to indicate the local crack development.

#### 2.2.3. Source Classification

Based on the features of the received signals, one can classify the source types into tensile cracking and friction [8]. Four parameters derived from the signal are relevant: rise time, amplitude, counts, and duration, as defined in Figure 5a. Based on these four parameters, two widely-applied features in source classification are RA value, which is the ratio of rise time and amplitude, and average frequency, which is the ratio of counts and duration [8]. Generally, signals from tensile cracking have a larger average frequency and a smaller RA value than those from friction [9,10]. These features were observed from two typical signals in our tests (Figure 5b). The counts are not marked in the typical signals because of their large number.

The two types of AE sources reflect different cracking behavior: tensile cracking is expected when the principle stress exceeds the tensile strength in concrete (crack opening); friction is more related to the crack displacement in the tangential direction of the crack surfaces (crack sliding). Therefore, we expect more signals from friction at a shear crack than at a flexural crack. This paper used the trend of decreasing average frequency and increasing RA value, which meant more signals from friction, to indicate the gradual opening from flexural cracks to shear cracks with increasing load.

### 2.3. Digital Image Correlation

At each load step, photos were taken of the DIC side. Figure 2 shows an example of one load step. For the four girders, a Canon 5DS R camera with 50.6 Megapixels resolution and a wide-angle lens (20 mm F1.4 fixed focus lens) were used on one side of the girder to capture the deformation in the web of the shear span. For HPZ02, HPZ03, and HPZ04, two extra cameras were used to capture the development of shear cracks in the web: a Canon 5D with 21 Megapixels with a macro lens (90 mm F/2.8 fixed focus lens) and a Sony A5000 (16–50 mm zoom lens at focal length of 46 mm) with 19.8 Megapixels. The camera-specimen distance for the Canon 5DS R was fixed to cover an area of 2590 × 1092 mm with a resulting pixel size equal to 0.31 mm. The Sony A5000 covered an area in the web equal to 980 × 602 mm with a pixel size of 0.19 mm. The measuring area of the Canon 5D was of 680 × 420 mm and the resulting pixel size was 0.13 mm. A scheme of the DIC setup with the areas of interest is shown in Figure 7.

In these experiments, a random speckle pattern was applied by a paint roller. The size of the speckles varied from 1 to 2 mm. DIC uses the random patterns to track and match subsets between two digital images [35]. Two LED (light-emitting diode) lights were used to provide constant illumination during the tests.

The DIC evaluation was performed with a publicly available Matlab code [36]. The code evaluated the degree of similarity between the subsets using a normalized cross-correlation criterion [36]. The difference in the locations resulted in the in-plane displacements. In this paper, the typical subset dimensions were 121 pixels for the Canon 5DS R and 71 pixels for the Canon 5D and the Sony A5000.

The displacement field obtained from the DIC Matlab code was further post-processed to obtain three different types of information which are of interest for practical engineers in monitoring post-tensioned bridge girders: (1) Crack pattern, (2) crack kinematics and aggregate interlock, and (3) angle of the compression field. In the following sections, the post-processing of the data is further explained.

First, the accuracy of the displacement measurement from the DIC analysis was evaluated with the LVDT readings. The wide-angle lens distortion was corrected following the procedure described in [37]. The locations of the mounting nodes of LVDT 5, LVDT 6, LVDT 7, LVDT 9 and LVDT18 were determined on the DIC side. Figure 8a shows the location of the nodes on HPZ04. At these locations, the relative displacement differences in the horizontal or vertical direction of the nodes were compared with the measurements of the LVDTs. The comparisons between the DIC and LVDT measurements are given in Figure 8b,d. This figure shows that the difference between the two methods is within 0.2 mm and 0.1 mm depending on the resolution of the camera and the lens. Considering that the crack pattern and the displacement field were not completely equal for both sides of the specimen, the difference was acceptable. Similar results were found for the remaining girders in the experimental program. In general, the accuracy of DIC is dependent on the quality of the pattern, the subset size, the resolution of the camera and the lens [38,39].

#### 2.3.1. Crack Pattern

The direct output of the DIC measurements was the displacement field of the measured surface, which was then converted into the strain distribution reflecting the crack pattern. From the raw output of equivalent strains, the cracks were identified and numbered.

The cracks obtained from the strain distribution were manually approximated to ten linear segments, from which the crack angles were calculated as shown in Figure 8a. The simplified crack pattern and crack angle were used for further analysis.

#### 2.3.2. Crack Kinematics and Aggregate Interlock

The DIC measurements allowed the determination of the crack kinematics in a detailed manner. The results of crack kinematics were divided into the displacement difference along the *x* and *y* direction, and the displacements along the crack profile.

For each crack, two measuring points on both sides of the crack profile were determined. The displacement difference in the *x*-direction was defined as *W* (crack opening in the longitudinal direction), and the displacement difference in the *y*-direction was defined as *∆* (crack opening in the transverse direction). An example of the measuring points is given in Figure 8a as *x_i_* and *y_i_*.

The crack kinematics in the global coordinate system were further converted to a local coordinate system along the simplified crack profile using the angle of the ten segments. As such, the local normal and tangential displacements were obtained. Detailed information about the algorithm can be found in [37].

The aggregate interlock stresses were computed using the normal and tangential displacements obtained directly from the DIC measurements as an input to Walraven’s formulation for aggregate interlock [40]. In Walraven’s model, the aggregates are simplified to rigid spheres and the cement matrix is an ideal plastic material. Slipping of the interface and crushing of the cement paste at the contact area generate the shear and normal stresses. The resulting forces are obtained by integrating the stresses along the crack profile. The normal and shear stresses are given as:(1)$$\left(\begin{array}{c} \sigma \\ \tau \end{array}\right)=\sigma_{pu}\left(\begin{array}{c} A_{y}+\mu A_{x}\\ A_{x}-\mu A_{y} \end{array}\right)$$
where *σ_pu_* is the compressive strength of the cement matrix, *μ* is the coefficient of friction, *A_x_*, *A_y_* are the projected contact areas between the surfaces of the aggregates and the cement matrix. The projected contact areas (*A_x_* and *A_y_*) depend on the magnitude of the normal (*n*) and tangential (*t*) displacements of the crack faces. Other influencing factors are the relative volume of aggregates and the distribution of the aggregate diameter.

#### 2.3.3. Angle of the Compression Field

The general web deformations were measured using the DIC results obtained from the cameras (Canon 5D, 90 mm lens and Sony A1000, 46 mm lens in Figure 7) responsible for local measurements. On the basis of these deformations, the angle of the compression field (*θ*) was calculated following Mohr’s circle using average strains as shown in Figure 9. It was assumed that the direction of the principal stresses and strains coincide. Thus, the angle of the compression field was estimated with the following equation:(2)$$\tan2\theta =\frac{\gamma_{xy}}{\varepsilon_{x}-\varepsilon_{y}}$$
where, *ε_x_* is the longitudinal strain in the web, *ε_y_* is the transverse tensile strain in the web, and *γ_xy_* is the shear strain in the web.

An algorithm was developed to select the area of interest and to calculate the average strains (*ε_x_*, *ε_y_*, and *γ_xy_*) from the displacement measurements of the DIC results. The normal strain in the *y*-direction (*ε_y_*) was calculated from the average displacements along the vertical line $\bar{BC}$ removing the outliers. For the normal strain in the *x*-direction (*ε_x_*), a linear strain distribution was assumed along the height direction with the strain at the top side of the web being zero. When AB is located at the bottom of the web, the assumed distribution result in $\varepsilon_{x}=\varepsilon_{x\bar{AB}}/2$ with $\varepsilon_{x\bar{AB}}$ as the average strains along the horizontal line $\bar{AB}$ located at the bottom of the web. The shear strain (*γ_xy_*) was computed as the change of the angle between the lines $\bar{AB}$ and $\bar{BC}$, thus $\gamma_{xy}=\alpha +\beta$, with *α* and *β* as shown in Figure 9.

With the measured angle of the compression field, we can estimate the number of stirrups activated using the NEN-EN 1992-1-1:2-005 [3], expression for the contribution of the stirrups (*V_Rd,s_)* as follows:(3)$$n_{stirrup}=\frac{\frac{A_{sw}}{s}zf_{yw}\cot\theta }{A_{sw}f_{yw}}=\frac{z}{s}\cot\theta$$
where *z* is the height of lever arm, *s* is the stirrup spacing, *θ* is the angle of the compression field, *A_sw_* is the area of the transverse reinforcement, and *f_yw_* is the yield strength of the transverse reinforcement.

## 3. Results

### 3.1. Tradional Measurement Results

Determination of the failure mode, based on the observations during tests, is presented in the companion paper [4]. A brief summary is: HPZ01 and HPZ02 failed by crushing of the compression strut between the load and the support (shear-compression failure); HPZ03 failed by local crushing of concrete in the top flange; and HPZ04 failed by crushing of the concrete compression field in the web of the girder.

In all experiments, the cracks gradually developed from sections with a higher sectional moment to sections with a lower sectional moment. Three types of cracks were observed: flexural cracks, which started from the bottom of the girder and developed upwards vertically, flexure-shear cracks, which were the inclined cracks developed from the previously defined flexural cracks, and shear–tension cracks, which were the inclined cracks that started in the web.

Table 1 gives an overview of the experimental results: *a* is the shear span, *a/d* is the shear span to depth ratio, *F_crack_* is the externally applied load at which the first flexural crack occurs, *F_fs_* is the externally applied load at which the first flexure-shear crack occurs, *F_st_* is the externally applied load at which the first shear–tension crack occurs, *F_fail_* is the externally applied load at which failure occurs, and the position where the critical shear crack crosses the mid-height of the web is indicated by *x_crit_* in Table 1.

Vertical web deformations measured by the LVDTs can help identify yielding of the stirrups. For example, Figure 10a shows the position of the stirrups and the vertical LVDTs in HPZ04. The results from the LVDTs are given in Figure 10b in which the yielding strain of the reinforcement (0.0023) is indicated with a dashed line. We can observe that strains at LVDTs 9–11 and 13 exceeded the yielding strain with increasing loading, but not for LVDT12. This could be because that LVDT12 was not in the shear cracking zone. An initial conclusion was that the stirrup next to LVDT12 was not yielded, while the other five stirrups near LVDTs 9–11 and 13 possibly yielded. However, the information from LVDTs was not sufficient to determine the number of stirrups that yielded. Further study on the angle of compression field is needed which is discussed in Section 3.2.3.

### 3.2. AE and DIC Measurements Results

#### 3.2.1. Cracking Load

Figure 11 shows the DIC visualization of the cracks with numbering from smaller to larger according to their opening sequence. First, flexural cracks developed (e.g., CR1 in HPZ01), then flexure-shear cracks (e.g., CR3 in HPZ01), then shear–tension cracks (e.g., CR7 in HPZ01), and finally the failure of the specimen occurred (shown on the right-hand side in Figure 11).

In this paper, AE and DIC determined the cracking load from the local cumulative AE events and the change of strain distribution, respectively. Table 2 lists the cracking load from AE and DIC in the four girders as *F_CRn_* the load at which crack number *n* developed, with the crack number following the numbering from Figure 11. A general observation in the four girders was that AE can detect cracking up to 100 kN earlier than DIC, in terms of externally applied load. Moreover, AE was sensitive to microcracking before a clear formation of a major crack was detected, giving a further advance detection of 50–100 kN.

Figure 12 and Figure 13 indicate the gradually opening of flexural cracks and shear cracks in HPZ04 in the measuring area of AE and DIC. For each load step, three types of plots are included: the first one is the estimated location of AE events that occurred in this load step, the second one shows the local cumulative AE events, and the third one is the incremental crack opening from DIC. The interval of each load step is 50 kN.

When microcracking and the first crack CR1 were detected by AE, DIC did not show a clear crack pattern. For the other cracks like CR2 and CR3, DIC can detect the cracking at the same load step as AE. The main reason was that small crack widths can be masked by the noise in the DIC measurements. The difference in crack width was also reflected by the local cumulative AE events, as in CR1 the value was less than 50, while in CR2 the value was over 100. This meant the opening of CR2 was wider with more released energy. The drawback of AE was a lower resolution compared to DIC due to the source localization error.

At a later stage, when more cracks opened, AE again detected cracks earlier than DIC. DIC was not able to detect the opening of CR5, further crack tip opening of CR2 and CR3 at 1450–1500 kN, when AE detected these activities (Figure 13a). AE can also detect opening of CR6 and CR9 earlier than DIC as can be seen in Figure 13b,c, respectively. However, AE was not sufficient to distinguish the actual crack pattern. More cracks between source and receiver delayed the wave arrival time and increased the source localization error, leading to unclear crack detection from AE at a later stage in the experiment.

Figure 14 shows the AE crack classification results. Two parameters, the RA value and average frequency, are plotted for AE events occurring in an earlier load step (1050–1100 kN) and later load step (1750–1800 kN). The former load step was when the first flexural crack CR1 was detected (see Figure 12b), and the latter was when the last shear crack CR10 was detected (see Figure 13d).

Comparing Figure 14a,b, more AE events tended to have lower average frequency and higher RA value in the later load step of 1750–1800 kN. This observation means that more friction events happened in the later load step when the shear crack CR10 opened. This result agreed with the expectation that friction between crack surfaces occurred more in a structural member with more shear cracks than flexural cracks. Figure 14c compares the change of mean RA value and mean average frequency obtained in each load step with the applied load level, showing the trend of decreasing average frequency and increasing RA value when the load increased.

As friction between two crack surfaces often occurred with aggregate interlock, more friction from AE crack classification can qualitatively show that more aggregate interlock was activated to carry shear with the increasing load.

#### 3.2.2. Aggregate Interlock

The aggregate interlock distribution was computed for all the experiments using the detailed results of crack kinematics and the crack profiles as introduced in Section 2.3.2. The results of aggregate interlock are given in Figure 15. For comparison, the loading scheme was also plotted in the same figure. The results for HPZ03 and HPZ04 considered only the last part of the loading scheme since flexure-shear and shear–tension cracks developed at this stage.

The general tendency shown in Figure 14 is that the aggregate interlock contribution was higher when the crack opening was smaller since larger contact areas were expected. The shear forces that were transferred through aggregate interlock of the cracks reduces with the increase of the applied load. The loss of aggregate interlock corresponded to an increase of the crack opening. For HPZ01, CR5 and CR6 merged into the critical shear crack which finally led to failure (see Figure 11a). A similar observation was found for HPZ02: CR4 and a secondary crack merged into the critical shear crack as shown in Figure 11b. Regarding HPZ03, the aggregate interlock could not be computed for the last load step due to a technical problem with the camera. However, the tendency indicates that there was a decrease of the contribution for CR4, CR5, and CR6. Figure 15d shows the results of the contribution of aggregate interlock for HPZ04. CR8, and CR10 present a decrease of aggregate interlock after the opening and development of CR8 at the load of 2145 kN. The shear stresses that were transferred through aggregate interlock in these cracks were almost zero at the moment before failure. CR9 presented an increase of aggregate interlock due to the compression of the cracked surface as a result of opening of the adjacent cracks. From these observations, we can conclude that aggregate interlock was not the governing shear-transfer mechanism. It became ineffective as the cracks widened and slid. The shear was transferred through the other actions, such as the stirrups and the direct strut from the load to the support.

#### 3.2.3. Angle of the Compression Field

The angle of the compression field *θ* was calculated from the strain measurements as a function of the load. Figure 16a shows the four areas that were analyzed with the DIC algorithm and Figure 16b presents the resulting angle as a function of time and the applied load for HPZ04. The general tendency was that before cracking the principal strain direction was approximately 45°. Then, after the development of the first inclined crack, the stiffness changes and the principal strain direction rotated to a lower angle. The angle tended to increase slightly when the load was held for a considerable amount of time as a result of the sustained loading. The results of the angle of the principal strain direction indicated that there was continuous rotation until failure, thus confirming that the angle of the compressive strut can be variable, as stated in the VATM.

Table 3 presents the results of the angle of the compression field (*θ_comp_*) at the moment before failure, computed as the average of at least three measured areas for all the experiments. The Table also includes the angle of the cracks (*θ_cracks_*) determined from the DIC results, as shown in Figure 17. The cracks that are within *d_web_* (height of the web) from the loading points are excluded in this analysis, as they are considered as flexural cracks. Moreover, the crack profiles were simplified into straight lines as was assumed in the VATM. From the figure, we can observe that the value of measured shear crack angles for HPZ04 varies from 17° to 31° with an average of 22° and that the angle of the compression field calculated from the DIC results is 18.5°. The angles were similar, which shows that at the moment before failure, aggregate interlock had a minimum effect and shear was carried by an equivalent truss as assumed in the VATM. If there was no aggregate interlock along the crack, the angle of the compression field would be equal to the crack angle. However, the angle of the compression field was lower because shear was being transfer through the crack due to aggregate interlock.

The measurement of DIC also supports the assumption that the actual angle of the compression field at failure is lower than the lower limit specified by NEN-EN 1992-1-1:2005 at 21.8°. With the measured angle of the compression field, we estimated the number of activated stirrups according to Equation (3). For HPZ04, we found five activated stirrups, with angle of compression of 18.5°, height of lever arm of 731.5 mm, and stirrup spacing of 400 mm. This result was also supported by the vertical web deformations measured by LVDTs (Figure 10 in Section 3.1). Following the same procedure, the estimated number of activated stirrups in HPZ01-03 was also five.

## 4. Discussion

### 4.1. AE and DIC Compared to Traditional Measurements

This paper presents the application of two advanced measuring techniques, i.e., AE and DIC, in the monitoring of existing prestressed concrete girders under laboratory conditions. AE and DIC can provide insights into relevant topics in the structural analysis including the cracking load, aggregate interlock and angle of compression field, which traditional displacement measurements cannot or can only partially provide.

The cracking load of the first flexural crack is an important indication which can be used to estimate the prestressing level of existing prestressed girders [19]. AE can detect the initiation of the first flexural crack at an earlier stage than the traditional displacement measurements due to its sensitivity to cracking, resulting in a lower cracking load and thus lower prestressing level. In the presented tests, the first flexural crack was detected at a smaller load level (50–100 kN less) compared to that obtained using the load-deflection relationship. The difference corresponds to about 10% of the applied load level. This advance may depend on the material and cross-sectional properties. In this study, the cracking load was determined per load step; an alternative approach is to check the change of the cumulative AE activities measured from the AE sensor in the vicinity of the flexural critical section. With the latter approach the cracking load may be even more accurately determined.

For the contribution of aggregate interlock, AE source classification can qualitatively indicate that more aggregate interlock was activated with increasing load, from the increasing number of AE events that were classified into friction. To quantify the aggregate interlock, DIC measurements based on the crack kinematics along the crack trajectory provide an effective tool in this study. The results show that aggregate interlock contributed to the shear-carrying capacity. However, it becomes ineffective when a crack presents a large opening and sliding in the tests.

DIC allowed us to determine the evolution of the compression field in the web of the girders during the test. With the increase of the sectional shear force, the angle of the compression field decreases constantly. This angle is an important parameter for the design and assessment for shear of concrete members with stirrups. The measured angle from DIC (18.1–20.1°) is lower than the lower limit specified in NEN-EN 1992-1-1:2005 [3], being 21.8°. The additional insight provided by DIC measurements enables the possibility of using a lower angle in evaluating the shear resistance of the existing prestressed concrete girders with a low shear reinforcement ratio. The lower angle of the compression field enables more stirrups in the web, thus increasing the actual estimated shear capacity. This observation is validated by the code evaluation of the experiments presented in the companion study [4].

### 4.2. Application of AE and DIC in Field Tests

The two measurement technologies have great potential to be applied in field tests of the existing concrete bridges. As the specimens are real structural components from existing bridges, the experiences that have been learned from the test program can be extended to actual field tests. These applications are further discussed in this section.

The design of the AE measurement, especially the sensor layout, needs to consider various aspects including the structural type, the material properties, the number of available sensors, and the size of the measuring area. For example, on-site measurements usually deal with large-size structures. With a limited number of AE sensors, it is suggested to first determine the interesting area, which could be the area with the largest moment or the area critical to shear cracking. In this way, we reduce the size of the measuring area and can install more AE sensors with less sensor spacing, which limits the influence of wave travel distance on the AE signals. In the next step, within the interesting measuring area, it is suggested to do a measurement on the wave transfer properties of the concrete medium including the wave speed and attenuation. This step is essential as these properties may vary with different structures and change over the course of the service life [41]. With the measured wave attenuation property, we can determine the maximum distance that allows signals to be detected by the sensors. In a previous study on reinforced concrete elements, a sensor spacing of 0.5 m showed to be sufficient for crack detection in reinforced concrete beams [29]. This sensor spacing was also found sufficient for monitoring the prestressed concrete girders in this paper. Only at the later stage near failure, the crack spacing got smaller in the web, it was then difficult for AE to distinguish those cracks. A solution could be to install AE sensors with less sensor spacing.

The design of the DIC setup in these experiments included three cameras with different focal lens. This is beneficial for the measurements of large-scale structures. The wide-angle lens camera allows us to capture a large measuring area, while the macro lens cameras provide detailed information in the critical area. Further improvements on the accuracy of the measurements could be achieved by using higher resolution cameras for all the measuring areas or improve the speckle pattern and size.

Although the tested specimens in this paper were of real size, they were still tested in the laboratory within a couple of hours (maximum two days). For long-term monitoring of a real bridge, a few more challenges need to be considered. First, environmental factors, such as rain and wind, may weaken the coupling between the AE sensor and the bridge surface. Under this circumstance, long-term AE monitoring requires effective protection of the sensors. For example, a protection box coving the sensors may be needed. DIC is affected by environmental factors, such as the change of lighting, which may affect the accuracy of the measurements. Other additional challenges are the preparation of the surface, the calibration of the cameras, and available camera locations. Then, as DIC needs a high-resolution camera, which is expensive, the DIC camera also requires protection. Moreover, the interesting area may not be accessible for DIC painting or AE sensor installation. In this case, for DIC, some researchers used the natural texture of the concrete (without painting patterns) [42]. While this paper shows promising results for the use of these techniques, validation through research studies in the field will be necessary before commercial deployment is possible.

## 5. Conclusions

This article deals with the results obtained in the shear tests on four post-tensioned prestressed concrete girders. These girders were heavily instrumented in the laboratory with traditional displacement sensors, acoustic emission (AE) sensors, and cameras for analysis of the photographs using digital image correlation (DIC).

AE could capture cracking before the traditional measurements and DIC, as AE is sensitive to cracking. In addition, the AE results have been used to identify the flexural cracking load of the girders, to derive the position of the cracks, and to identify the source of cracking (tensile cracking or friction).

The DIC algorithm gives further insight in the structural behavior of post-tensioned girders failing in shear. The visualization of the crack pattern helps in identifying the sequence of crack development. Studying the crack kinematics gives insight in the contribution of aggregate interlock as a shear-carrying mechanism. Moreover, analyzing the strains with Mohr’s circle shows that the compression field rotates from 45° before cracking down to 18.1–20.1° at failure.

For the assessment of shear-critical post-tensioned girders, insights from AE and DIC help to better understand the shear-transfer mechanisms, including aggregate interlock and the contribution of the stirrups. The results confirm that aggregate interlock contributes to the shear capacity, but with larger crack openings, other mechanisms take over, which could be the shear-carrying capacity of the stirrups or the direct compression strut from the load to the support. The contribution of stirrups can be estimated from the DIC measurement of the angle of compression field. The angle of the compression field measured by DIC is smaller than that prescribed in the design code based on the VATM.

Ultimately, the combination of AE and DIC gives valuable insights in the cracking and shear-carrying behavior of prestressed concrete girders. This method could also be used for long-term monitoring in the field, but further studies would be necessary to check the environmental influences on the functioning of the sensors.

## Figures and Tables

**Figure 1 sensors-20-05622-f001:**
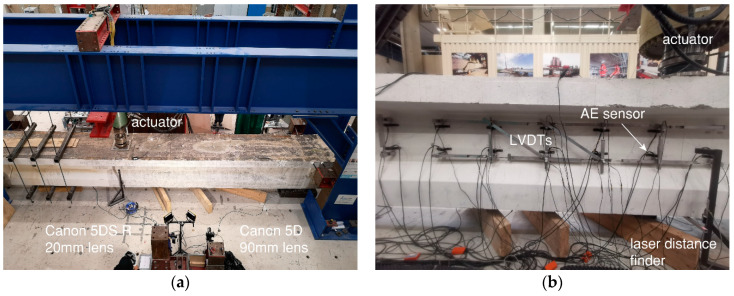
Overview of test setup: (**a**) At DIC side, and (**b**) at sensor side.

**Figure 2 sensors-20-05622-f002:**
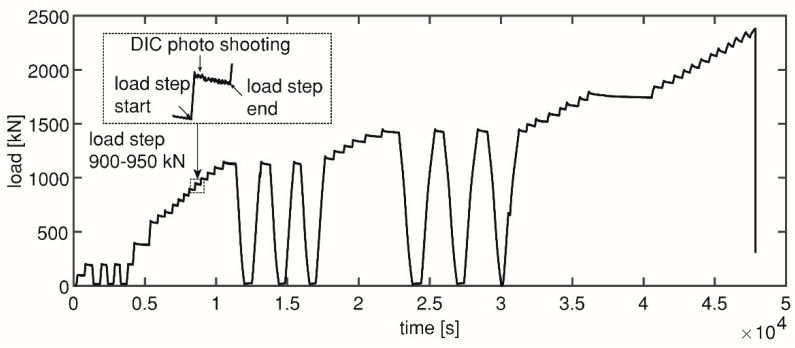
Loading history of HPZ04.

**Figure 3 sensors-20-05622-f003:**
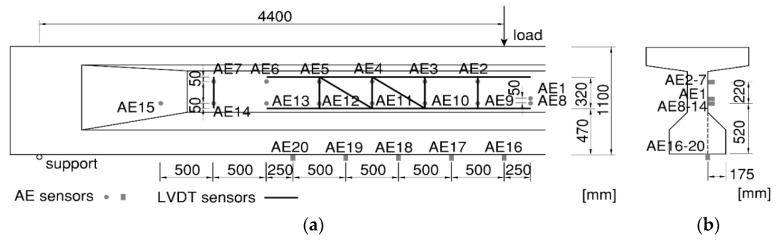
Sensor plan for beam HPZ04 showing LVDTs and AE sensors: (**a**) Side view; (**b**) cross-section.

**Figure 4 sensors-20-05622-f004:**
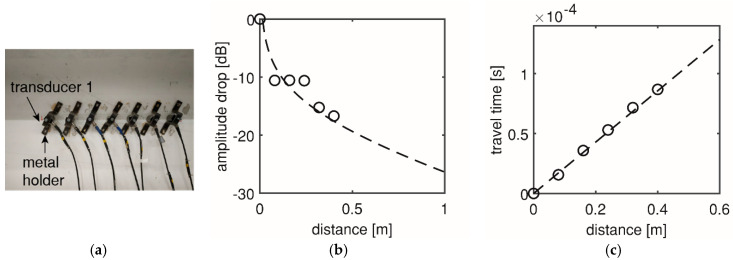
Baseline measurement in HPZ03: (**a**) Setup, (**b**) amplitude drop, and (**c**) wave speed.

**Figure 5 sensors-20-05622-f005:**
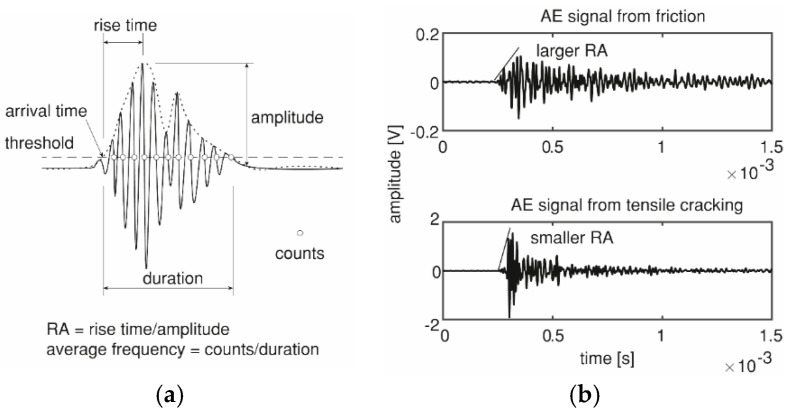
(**a**) Illustration of AE parameters and (**b**) AE signals from friction and tensile cracking.

**Figure 6 sensors-20-05622-f006:**
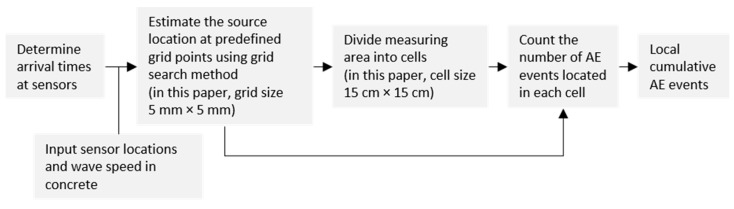
Approach of calculating local cumulative AE events.

**Figure 7 sensors-20-05622-f007:**
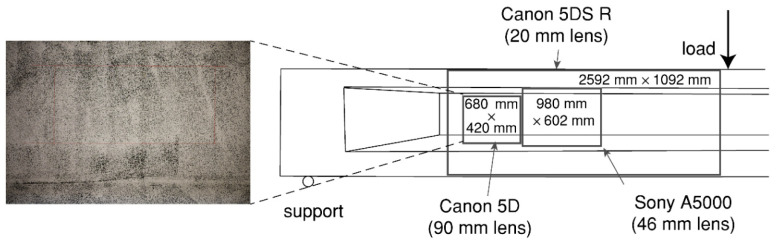
DIC setup for beam HZP04 showing the random pattern, and the position of the 20 mm wide angle lens, the 90 mm lens and the SONY A5000 camera.

**Figure 8 sensors-20-05622-f008:**
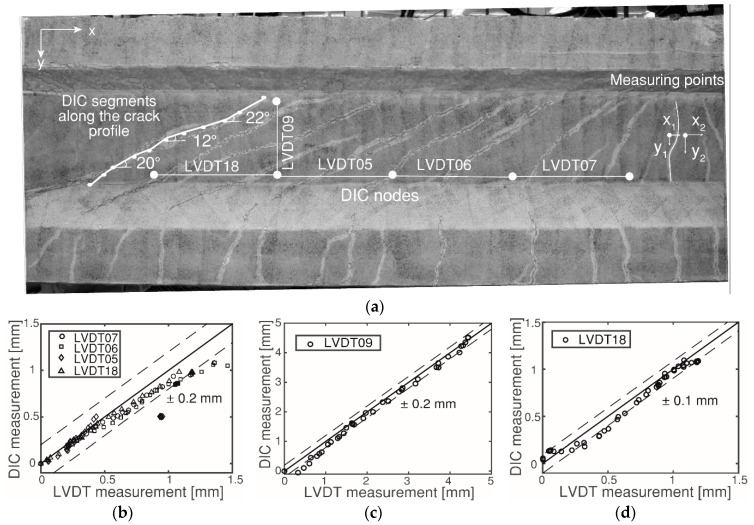
Calibration of DIC with LVDT in HPZ04: (**a**) location of DIC nodes and LVDTs 05–07, LVDT09, and LVDT18, together with the segmental division in one crack; (**b**) LVDTs 05–07 and LVDT18 measurements and DIC with Canon 5DS R (20 mm lens); (**c**) LVDT09 measurement and DIC with Sony A5000; and (**d**) LVDT18 measurement and DIC with Canon 5D (90 mm lens).

**Figure 9 sensors-20-05622-f009:**
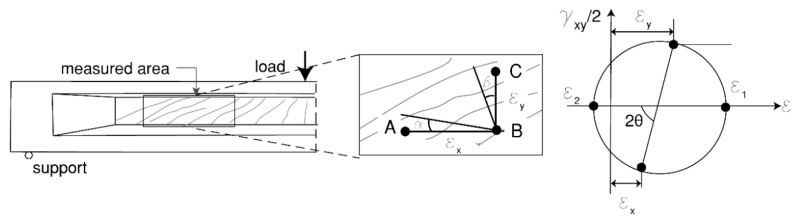
Scheme of measured area for HPZ04 and Mohr’s circle for average strains.

**Figure 10 sensors-20-05622-f010:**
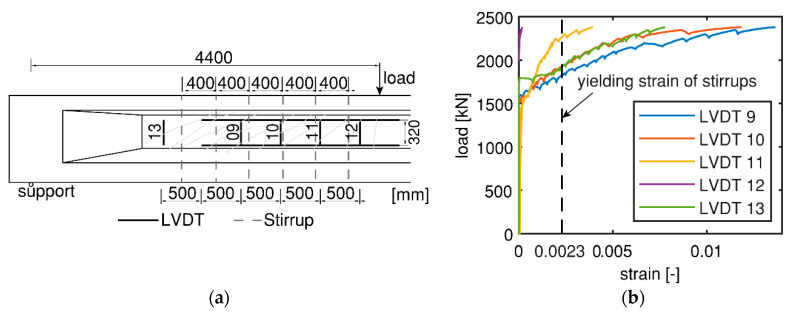
Measurement of yielding stirrups from LVDTs for HPZ04: (**a**) Position of the stirrups and LVDTs; and (**b**) Vertical deformation results of LVDTs.

**Figure 11 sensors-20-05622-f011:**
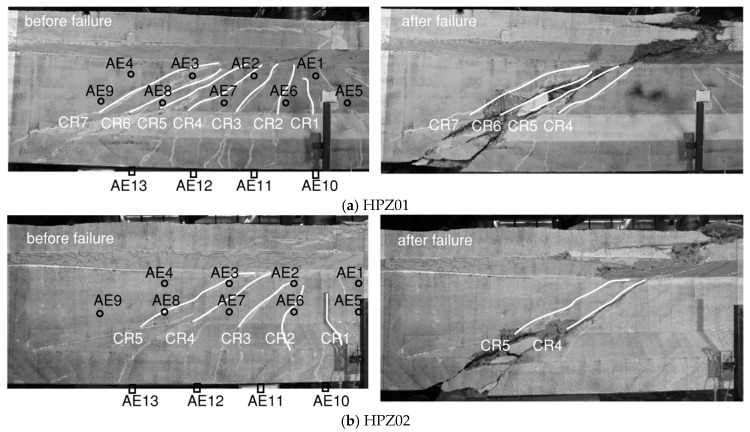
Crack patterns in the four girders from DIC results before and after failure, with the locations of AE sensors: (**a**) HPZ01, (**b**) HPZ02, (**c**) HPZ03, (**d**) HPZ04.

**Figure 12 sensors-20-05622-f012:**
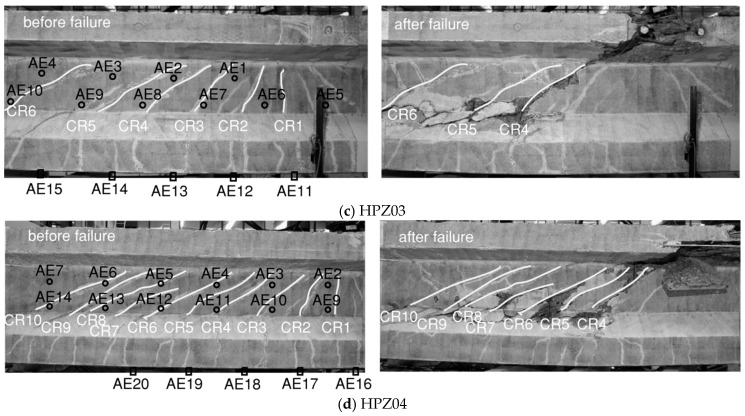
Opening of flexural cracks in HPZ04: (**a**) Microcracking; (**b**) CR1; (**c**) CR2; and (**d**) CR3. At each subfigure, from top to bottom: the first plot is the estimated location of AE events, the second plot shows the local cumulative AE events, and the third plot shows the incremental crack opening from DIC. The DIC results are edited to help the reader identify the important cracks.

**Figure 13 sensors-20-05622-f013:**
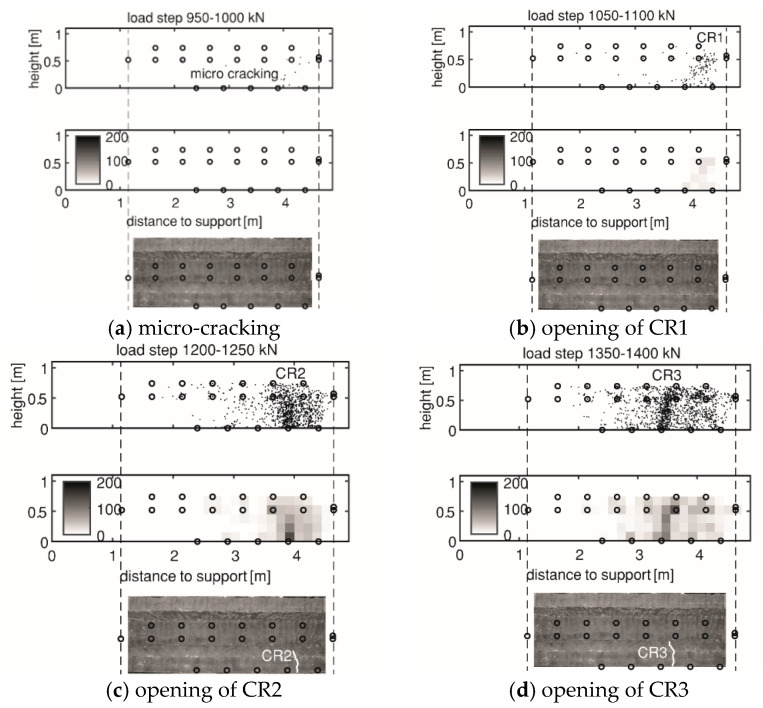
Opening of shear cracks in HPZ04: (**a**) CR5; (**b**) CR6; (**c**) CR9; and (**d**) CR10. At each subfigure, from top to bottom: the first plot is the estimated location of AE events, the second plot shows the local cumulative AE events, and the third plot shows the incremental crack opening from DIC. The DIC results are edited to help the reader identify the important cracks.

**Figure 14 sensors-20-05622-f014:**
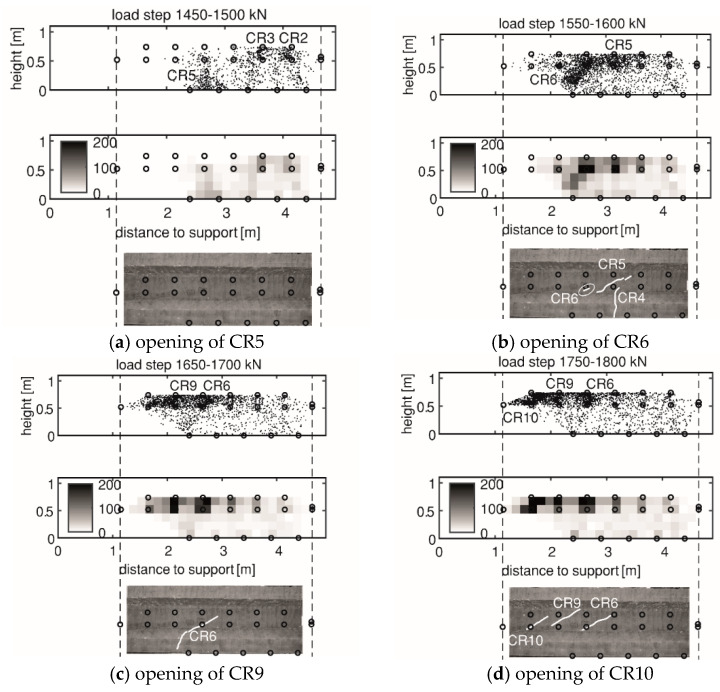
(**a**,**b**): Average frequency and RA value of each AE event at load steps 1050–1100 kN and 1750–1800 kN, respectively, and (**c**): Changing of mean average frequency and mean RA value with loading.

**Figure 15 sensors-20-05622-f015:**
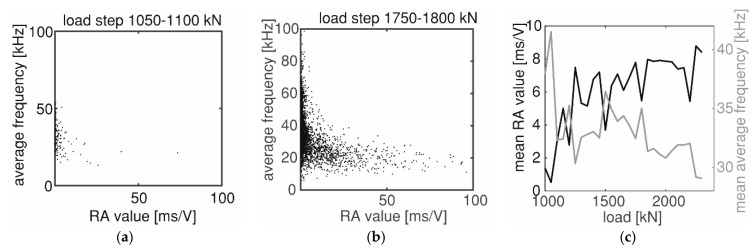
Aggregate interlock contribution during loading analyzed over time for all the experiments: (**a**) HPZ01; (**b**) HPZ02; (**c**) HPZ03; and (**d**) HPZ04. * indicates the critical shear cracks.

**Figure 16 sensors-20-05622-f016:**
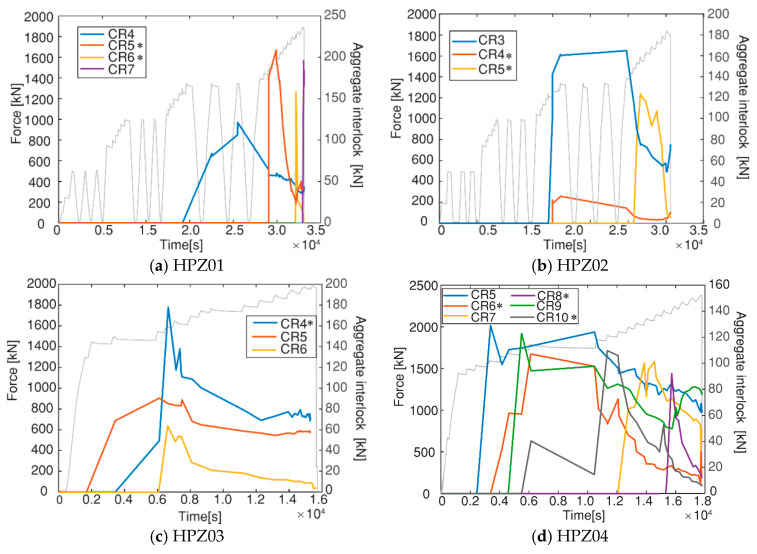
(**a**) Evaluated areas for HPZ04, and (**b**) Rotation of the principal strain direction during loading for HPZ04.

**Figure 17 sensors-20-05622-f017:**
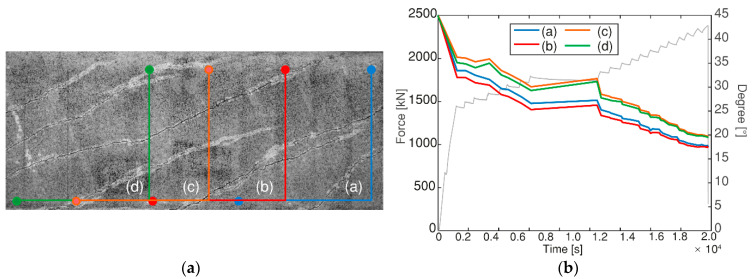
Crack pattern and crack angle from DIC results of HPZ04.

**Table 1 sensors-20-05622-t001:** Overview of experimental results.

Girder No.	*a* (mm)	*a/d*	*F_crack_* (kN)	*F_fs_* (kN)	*F_st_* (kN)	*F_fail_* (kN)	*x_crit_* (mm)
HPZ01	2903	3.6	965	1344	1480	1893	1828
HPZ02	2903	3.6	1001	1299	1350	1849	1873
HPZ03	4400	4.9	1050	1250	1600	1990	3460
HPZ04	4400	4.9	1100	1450	1750	2380	2832

**Table 2 sensors-20-05622-t002:** Overview of the cracking load from DIC and AE in the four girders.

Girder No.	Technique	*F_CR1_* (kN)	*F_CR2_* (kN)	*F_CR3_* (kN)	*F_CR4_* (kN)	*F_CR5_* (kN)	*F_CR6_* (kN)	*F_CR7_* (kN)	*F_CR8_* (kN)	*F_CR9_* (kN)	*F_CR10_* (kN)
HPZ01	DIC	1100 ^1^	1200	1350	1350	1470 *	1790	1880			
	AE	850/950 ^2^	1150	1300	1350	- ^3^	-	-			
HPZ02	DIC	1100	1220	1300	1300 *	1550					
	AE	950/1000	1245	1300	1300	-					
HPZ03	DIC	1150	1050	1250	1550	1400	1600 *				
	AE	1200	950/1000	1250	1400	-	-				
HPZ04	DIC	1150	1250	1400	1600	1535	1600 *	1830	2090	1750	1790
	AE	1000/1100	1250	1400	-	1500	1600	-	-	1700	1800

^1^ The crack could have been recognized earlier, but the steel frame to support the laser blocked the view of the area of interest. ^2^ The former was the microcracking load, and the latter was the cracking load. The same applies to other entries with two values. ^3^ AE could not accurately locate the crack. The same applies to other dashes. * The crack was the first shear–tension crack.

**Table 3 sensors-20-05622-t003:** Angle of compression field *θcomp* at the moment before failure for all the experiments from different measurement approaches. *θcomp* is the angle of the compression field, and *θcracks* is the angle of the inclined cracks.

	*θ_comp_* [°]	*θ_cracks_* [°]
HPZ01	18.1	22–34
HPZ02	20.1	25–33
HPZ03	19.9	22–31
HPZ04	18.5	17–31

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
