# Peer review of "Monitoring Shear Behavior of Prestressed Concrete Bridge Girders Using Acoustic Emission and Digital Image Correlation"

_sensors, 2020, doi:10.3390/s20195622_

Round 1

Reviewer 1 Report

1、In the section of 2.1.3, I am very curious about the specific numbers of AE sensor? Is it necessary to use so many AE sensors?

2、In section 2.2.2, I would like to see your mathematical formulas reasoning process of localization methods with respect to the sound source location, or the flow-process diagram of location technology.

3、In section 2.2.3, what algorithm do you use to classify sound sources? Please briefly describe.

4、In section 2.3, is the camera fixed or scanned for taking pictures? If it is a fixed photo, why do you only take a certain fixed area?

5、In Section 2.3, please explain what algorithm is used for post-processing of data?

Reviewer 2 Report

Reviewer report Sensors

Paper: Monitoring shear behavior of prestressed concrete bridge girders using acoustic emission and digital image correlation

General Comment

Although I do not consider myself an expert in measuring and/or assessing shear behavior on concrete, I work with AE and DIC technique on other materials.

As general comment, I found the research work of the authors related to the laboratory experiment on the shear behavior of prestressed concrete bridge well done and wide as:

  • They exhaustively explained the preliminary needed tests to check the experimental setup and the AE features (e.g. wave speed, attenuation and background noise). Moreover, this experimental approach, as it has been tested to be effective in the laboratory, in the future, can be applied on case study monitoring in the field as the specimens came already from real structural components from existing bridges.
  • Then, comparing the 2 innovative NDT techniques of the AE and the DIC, during the data analysis (especially in the sub-section: cracking load) they used exhaustive visual tools (e.g. plot and B&W mapping) to make comprehensible to the reader the cracking load formation patterns.
  • I found interesting the further analysis on the classification of AE crack results
  • In conclusion, the authors were able to demonstrate that AE technique can detect the initiation of the first flexural crack at earlier stage than the traditional displacement measurements due to its sensitivity to cracking, detectable at smaller load level 50-100 kN lower than DIC.
  • Finally, I found interesting, in the discussion section, the arguments to implement in reality a long term monitoring campaign in the field.

Therefore, I suggest the publication of this work after that minor revisions have been solved.

Minor Comments

Line 88: explanation of the acronym LVDT linear variable displacement transducer needs to be added the first time it is used.

line 133: Explains better why, for each sample a different experimental setup, with different load/unload cycles numbers was used

Line 142: missing reference source

Line 147: change in : “can increase the measuring area”

Line 150-151: “which was proven to be effective in”

Lines 154, 161 and 167: errors in references that are missing

Line 167: I do not understand the label (b) here in the text. I suggest to revise the order in which the baseline measurement are presented in the manuscript, following the same order as in Figure 4 i.e. setup, wave speed findings and the amplitude drop measures findings.

Line 174: again here I suggest to re-labelling figure 5 as the authors quoted first Figure 5b this will become figure 5a and viceversa with figure 5a that becomes 5b.

Lines 180-194. It is not clear the differences, between the sensor location distance (0,5m), The grid (5mm) and the cell dimension 15 cm. Please re-organize the section to make them clearer.

Table 2: improve the layout reducing the font size of the header

Figure 12a: is the DIC not able to detect the incremental crack opening (bottom plot) for CR2, CR3 and CR5? Please briefly comments the results in Figure 12 too.

Figure 14- the label and unit of the right y axis is missing.

Line 493: add a space between “Equation” and “(3)”

Round 2

Reviewer 1 Report

All questions have been revised,I think it can be accepted for publication.